# Three-dimensional character of the deformation twin in magnesium

Y. Liu [1,2], P.Z. Tang[1], M.Y. Gong[2,3], R.J. McCabe[2], J. Wang[3] & C.N. Tomé[2]

Deformation twins are three-dimensional domains, traditionally viewed as ellipsoids because of their two-dimensional lenticular sections. In this work, we performed statistical analysis of twin shapes viewing along three orthogonal directions: the 'dark side' (DS) view along the twin shear direction ($\eta_1$), the twinning plane normal (TPN) view ($k_1$) and the 'bright side' (BS) view along the direction $\lambda(=k_1 \times \eta_1)$. Our electron back-scatter diffraction results show that twins in the DS and BS views normally exhibit a lenticular shape, whereas they show an irregular shape in the TPN view. Moreover, the findings in the TPN view revealed that twins grow faster along $\lambda$ the lateral direction than along $\eta_1$ the forward propagation direction at the initial stages of twin growth. These twin sections are irregular, indicating that growth is locally controlled and the overall shape is not perfectly ellipsoidal. We explain these findings using atomistic models, and ascribe them to differences in the mobility of the edge and screw components of the twinning dislocations.

[1] State Key Lab of Metal Matrix Composites, School of Materials Science and Engineering, Shanghai Jiao Tong University, Shanghai 200240, China.
[2] Materials Science and Technology Division, Los Alamos National Laboratory, Los Alamos, NM 87545, USA. [3] Department of Mechanical & Materials Engineering, University of Nebraska-Lincoln, Lincoln, NE 68583, USA. Correspondence and requests for materials should be addressed to Y.L. (email: yliu23@sjtu.edu.cn) or to C.N.T. (email: tome@lanl.gov)

Magnesium (Mg) and Mg alloys with hexagonal closed packed (HCP) structure have potential applications as structural components and in the transportation industry due to their high strength-to-weight ratios. Understanding the mechanical properties and underlying plasticity mechanisms is essential for optimizing material processing and improving performance[1–3]. HCP metals deform plastically via dislocation slip and twinning at room temperature. The easy slip associated with $\langle a \rangle$ dislocations on either the (0001) basal plane or the $\{10\bar{1}0\}$ prismatic plane does not accommodate deformation along $c$-axis. On the other hand, $\langle c + a \rangle$ dislocations on nonbasal planes that can contribute to deformation along the c-axis are difficult to activate at room temperature due to high lattice friction[4–6]. As a consequence, twinning represents an alternative shear mechanism that accommodates plastic deformation along the $c$-axis[7–9] and so influences the ductility and formability of HCP metals[10–13]. In the case of Mg and its alloys, $\{01\bar{1}2\}$ extension twins are most commonly observed (Fig. 1a)[14–16]. Twinning that involves three sequential processes: nucleation, propagation, and growth, are associated with the formation and migration of twin boundaries (TBs). This work focuses on the twin morphology at the propagation stage, defined as the twin having partially or fully traversed the grain, but before it starts growing in thickness after being arrested at the opposite grain boundaries (GBs). The goal here is to derive basic knowledge of the motion mechanisms of TBs, essential to understand mechanical behavior dominated by twinning processes and interactions[17–20].

Because of the three-dimensional (3D) nature of twin domains, twin growth involves motion of TBs along directions that are in the $k_1$ plane and perpendicular to the $k_1$ plane. We analyze this 3D growth by characterizing twins along three directions: the 'dark side' (DS) view along the twin shear direction ($\eta_1$), the twinning plane normal (TPN) view ($k_1$) and the 'bright side' (BS) view along the direction $\lambda (= k_1 \times \eta_1)$. They are identified with arrows in an HCP structure (Fig. 1a), a 3D schematic (Fig. 1b), and 2D schematic twin sections (Fig. 1c, d, e). Until now —with a few recent exceptions[21–23]—most studies of these 3D domains have focused only on sections revealing the BS view along the direction $\lambda$. Assuming that nucleation takes place at a GB, a twin grows through successive nucleation and propagation of twinning disconnections (terraces). Their passage builds-up a 3D domain (Fig. 1b) where a finite shear concentrates. These twin domains appear to be polyhedral and bounded by low-energy steps and facets formed by the propagation of twinning dislocations (TDs)[24,25].

Extensive studies have been conducted to reveal the characteristics of twins in the BS view, showing normal and forward TBs, as illustrated in Fig. 1c. The forward TBs exhibit semi-coherent $\{01\bar{1}0\}$ prismatic $\|$ (0002) basal (referred to as BP/PB) facets observed experimentally in Mg[26,27], Cobalt[28], Titanium[29], and also in atomistic simulations of Mg[30–32]. The normal TB commonly exhibits a serrated appearance composed of $\{01\bar{1}2\}$ coherent twin boundary (CTB) facets and BP/PB steps. By comparison, there is limited knowledge regarding the lateral growth of the twin, partly owing to the challenges to observe and characterize the TBs in the DS view (Fig. 1d). Our recent work has characterized the lateral TBs at an atomic level, and found that they are composed of CTBs and semicoherent twist $\{2\bar{1}\bar{1}0\}$ Prismatic$\|$Prismatic (TPP) steps[24,33].

Despite these atomic level studies showing semicoherent BP/PB and TPP boundaries in BS and DS views, respectively, some important questions remain unanswered. First, whether the kinematics of twin growth are comparable for BS and DS sections. Second, the effect of the different types of TDs and semicoherent facets on the mobility of a twin remains a mystery. These issues can be addressed by characterizing the twin shape in the TPN view (along $k_1$), because this view provides the dimensions of a twin in the forward and lateral directions (Fig. 1e). In this study we characterized multiple $\{01\bar{1}2\}$ twin shapes along $\eta_1$, $k_1$, and $\lambda$

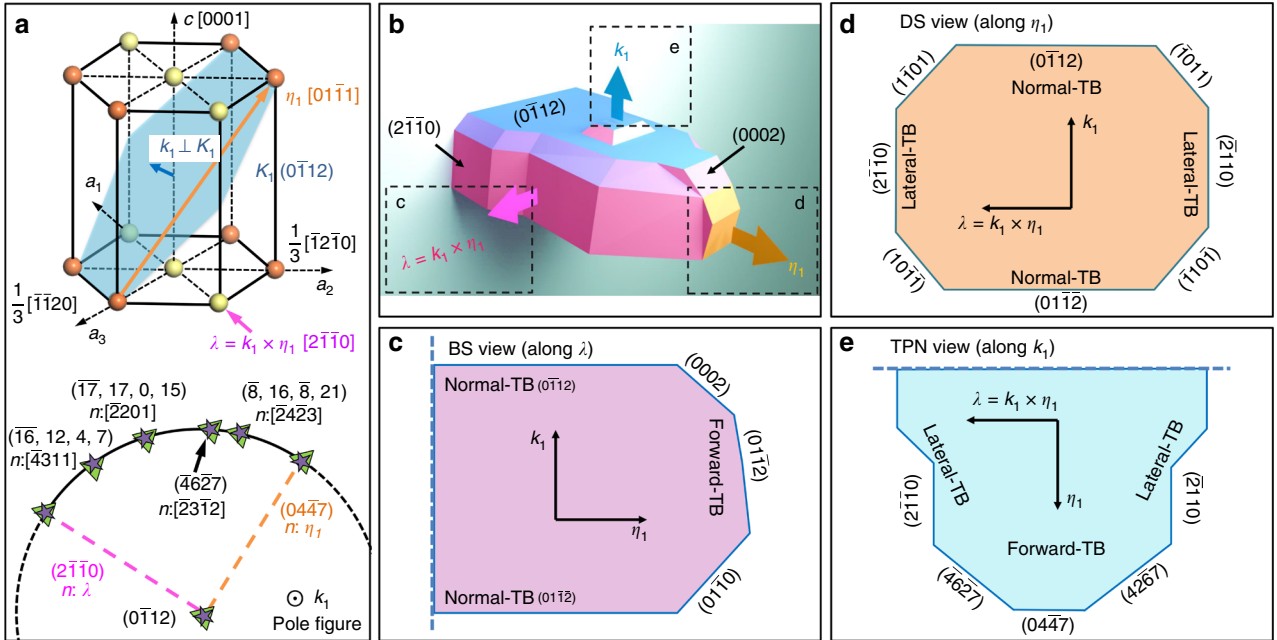

**Fig. 1** The three-dimensional view of twinning associated boundaries. **a** Coordinates for hexagonal closed packed (HCP) structure and the pole figure of $(0\bar{1}12)$ twinning plane. **b** Three-dimensional twin schematic with reported twin facets defining twin boundaries (TBs). **c** 'Bright side' (BS) view of the twin along $\lambda = k_1 \times \eta_1$ showing normal-TB and forward-TB. **d** 'Dark side' (DS) view of the twin along $\eta_1$ showing normal-TB and lateral-TB. **e** Twin plane normal (TPN) view of the twin showing forward-TB and serrated lateral-TB. $(0\bar{1}12)$ coherent TBs can be observed in both, BS view and DS view, but not in TPN view. The possible serrations may be $(\bar{4}6\bar{2}7)$ and $(4\bar{2}\bar{6}7)$ based on the pole figure and stereogram analysis in Fig. 1a, Supplementary Fig. 1, and Supplementary Fig. 2

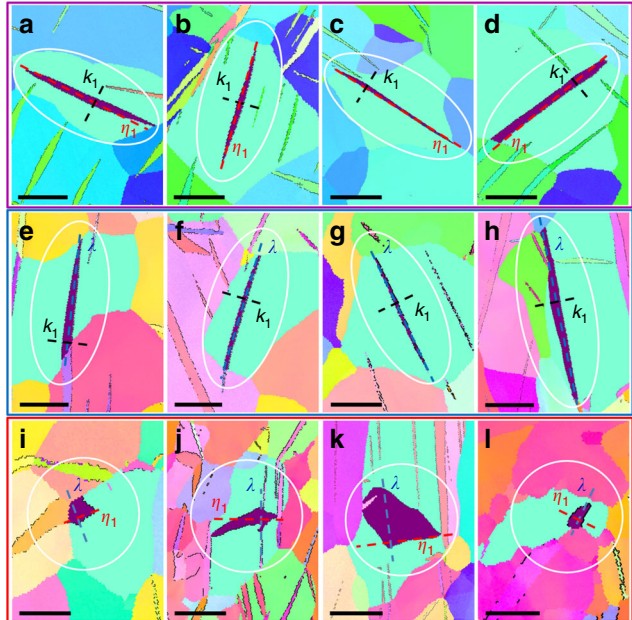

**Fig. 2** Twin shapes in the BS (lenticular), DS (lenticular), and TPN view (irregular). **a–d** Typical twin sections in the BS view with short axis along $k_1$ and long axis along $\eta_1$. **a** Scale bar, 45 μm. **b** Scale bar, 40 μm. **c** Scale bar, 75 μm. **d** Scale bar, 50 μm. **e–h** Typical twin sections in the DS view with short axis along $k_1$ and long axis along $\lambda$. **e** Scale bar, 45 μm. **f** Scale bar, 40 μm. **g** Scale bar, 50 μm. **h** Scale bar, 70 μm. **i–l** Typical twin sections in the TPN view showing the position of the $\eta_1$ and $\lambda$ axes. **i** Scale bar, 50 μm. **j** Scale bar, 85 μm. **k** Scale bar, 90 μm. **l** Scale bar, 60 μm. The highly irregular shape suggesting low-surface tension, and mostly lateral propagation of the twin in case (**l**). The detailed analysis to determine length of axis, twin relation and crystallographic directions can be found in Fig. 3, Supplementary Fig. 3, and Supplementary Fig. 5. The longest length of each axis in each section was used for the statistical information reported in Fig. 4

directions using sections perpendicular to those directions. The statistical electron back-scatter diffraction (EBSD) results show that twins exhibit a lenticular shape in DS and BS views, but an irregular shape in TPN view. Experimental results also show that twins grow faster in the direction $\lambda$ (associated with migration of lateral TBs) than in the direction $\eta_1$ (associated with migration of forward TBs) at the initial stages of twin growth, before intersection with GBs that arrest the forward and lateral propagation.

## Results

**Twin shapes in the BS, DS, and TPN view.** Figure 1a, Supplementary Fig. 1, and Supplementary Note 1 show the $(01\bar{1}2)$ twinning plane pole figure and stereogram when observing in the TPN view. This analysis is critical to identify twin relations (Supplementary Fig. 2) or possible low-index crystallographic facets (such as $(42\bar{6}7)$ and $(\bar{4}6\bar{2}\bar{7})$ in Fig. 1e). As shown in Fig. 2, EBSD results reveal lenticular shapes in both the BS view (Fig. 2a–d) and the DS view (Fig. 2e–h), but an irregular shape in the TPN view (Fig. 2i–l). The intersecting traces of particular crystallographic planes or directions (such as traces along $\lambda$, $\eta_1$, and $k_1$) can be identified by pole figure analysis (Supplementary Fig. 3 illustrates the details of this analysis method). Therefore, as shown in Fig. 2 (more detailed information in Supplementary Fig. 4 and Supplementary Fig. 5), we find that twin shapes in the BS view are much narrower along $k_1$ than along $\eta_1$, and twin shapes in the DS view are much narrower along $k_1$ than along $\lambda$. By comparison, the twin in the TPN view has an irregular shape

and no obvious correlation between $\lambda$ and $\eta_1$. This irregular shape is likely due to the different types of facets and TDs that affect the mobility of TBs. Based on this qualitative evidence, we performed a significant amount of EBSD experiments to explore the characteristics of twin propagation and growth along the $\lambda$, $\eta_1$, and $k_1$ orthogonal directions.

**Selection criteria and statistics of qualified twins.** The criteria to select the qualified twins are the following. First, the deviation between the viewing axis $\lambda$, $\eta_1$, or $k_1$ of each twin has to be less than 5°. Second, if there exists a deviation, the measured lengths are adjusted using the pole figures of each twin as shown in Fig. 3. Third, only twins with none or minimum overlap with GBs are considered. The rationale is that when a large portion of the TB is shared with a GB, the twin shape is greatly determined by grain morphology. Therefore, grain sizes are normally greater than the twin sizes reported here. Figure 4 presents the statistical results of twin aspect ratios for the BS, DS, and TPN view. We classified twins in three types based on their junctions at GBs as they appear in the EBSD sections. They are Type 0 if neither side of the twin is connected to a GB, Type 1 if only one side is connected to a GB and Type 2 if both sides are connected to GBs. Based on these criteria, 276 twins are analyzed. As shown in Fig. 4a, the number of qualified BS twins is relevant for the three types, with prevalence of type 2. Twins in the DS view are overwhelmingly type 2, but practically zero number of type 0. In what concerns twins in the TPN view, they are type 1 and type 2 in about equal numbers, indicating that propagation always starts at a GB. This evidence suggests that the lateral expansion of twins is relatively easy, since most of the DS view $(\lambda - k_1)$ extend from side to side of the grain (type 2), compared to the BS view $(\eta_1 - k_1)$.

**Irregular twin shapes in the TPN view.** Statistical results of the maximum lengths measured in the BS view sections ($\eta_1 - k_1$, in pink) and DS view sections ($\lambda - k_1$, in blue) and TPN view sections ($\lambda - \eta_1$, in red), plus their ratio ($\frac{k_1}{\eta_1}$, $\frac{k_1}{\lambda}$ and $\frac{\lambda}{\eta_1}$) are illustrated in Fig. 4b–d. These results confirm the lenticular twin shape in the BS and DS views and the irregular twin shape in the TPN view. In all cases, statistical results show that whatever the maximum length along $\lambda$ and $\eta_1$ is, the maximum length along $k_1$ is much smaller. It emphasizes that lateral and forward propagation are faster than the growth normal to the $k_1$ plane, which results in lenticular shapes for both BS and DS view twin sections. In contrast, the statistical correlation between the maximum lengths along $\lambda$ and $\eta_1$ manifests the irregular twin shapes associated with the TPN view in general.

**Faster lateral propagation after a twin nucleates.** As shown in Fig. 2i–l, neither $\lambda$ nor $\eta_1$ is along the long or the short axis of the twin section in the TPN view, and the expansion along $\lambda$ or along $\eta_1$ is not confined by a GB in most cases. The statistical distribution of the $\frac{\lambda}{\eta_1}$ ratio of each twin shown in Fig. 4d suggests that, after a twin nucleates, it tends to propagate faster laterally along $\lambda$ than along the shear direction $\eta_1$. Such behavior holds up to $\lambda$ ~40 μm, above which the length $\eta_1$ appears to be systematically larger than $\lambda$. However, the scarcity and dispersion of the results can not guarantee a firm conclusion, and we speculate that at this point the propagation of the twin expanding is affected by grain boundaries or by other twins. This finding not only underlines the importance of the lateral expansion (DS view) on three-dimensional twin growth, but also poses an important question: the motion mechanisms that the lateral facets in the DS

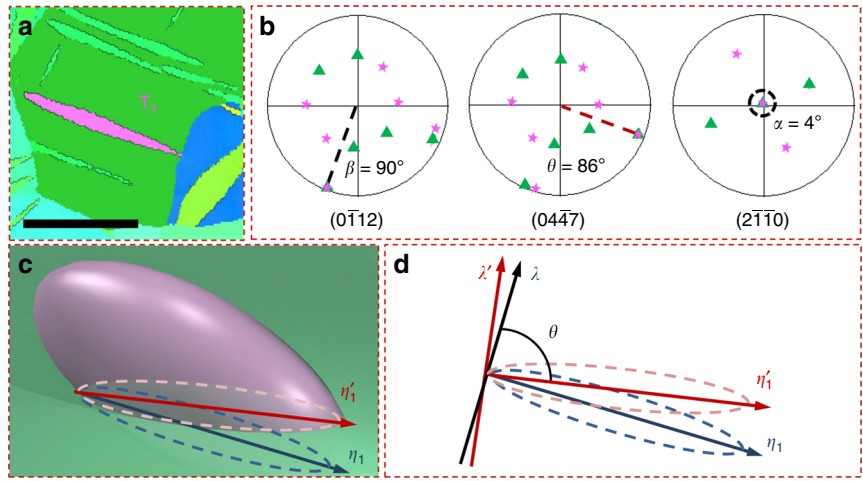

**Fig. 3** Showcase of length determination along crystallographic directions. **a** A deformation twin $T_1$ (color in pink) from BS view. Scale bar, 60 μm. **b** Corresponding pole figures of the twin and matrix (color in green). The pole figures show deviation angles $\alpha = 4°$ from $\{2\bar{1}\bar{1}0\}$ and $\theta = 86°$ from $\{40\bar{4}7\}$ (plane normal to $<10\bar{1}1>$) pole figures. **c, d** Illustration of the projection plane on electron back-scatter diffraction (EBSD) sections and the measured length correction along $\eta_1$ based on the deviation angles quantified by pole figures

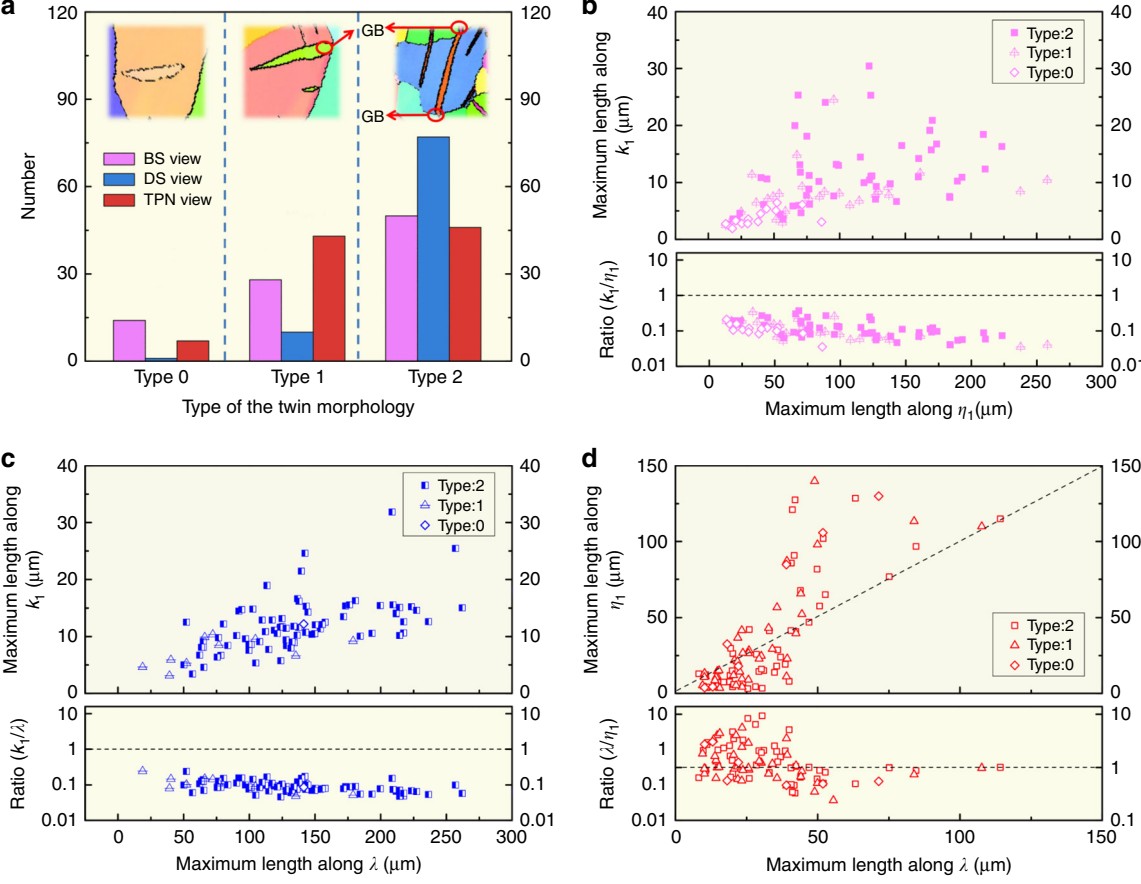

**Fig. 4** Statistical results confirm irregular twin shapes in the TPN view. **a** Number of twins that are categorized as Type 0 (no grain boundary (GB) junction), Type 1 (one GB junction), and Type 2 (two GB junctions) corresponding to sections in the BS, DS, and TPN view. Statistical distribution and relationship of: **b** $k_1$ versus the long axis $\eta_1$ for BS twin sections; **c** $k_1$ versus the long axis $\lambda$ for DS twin sections; **d** $\eta_1$ versus $\lambda$ for TPN twin sections. In all cases the maximum value of the axes in each twin section was selected for the plots. The statistical results confirm the lenticular twin shape in the BS and DS view, and the irregular twin shape in the TPN view. **d** The fact that $\eta_1 < \lambda$ suggests that lateral growth is faster when $\lambda < 40$ μm

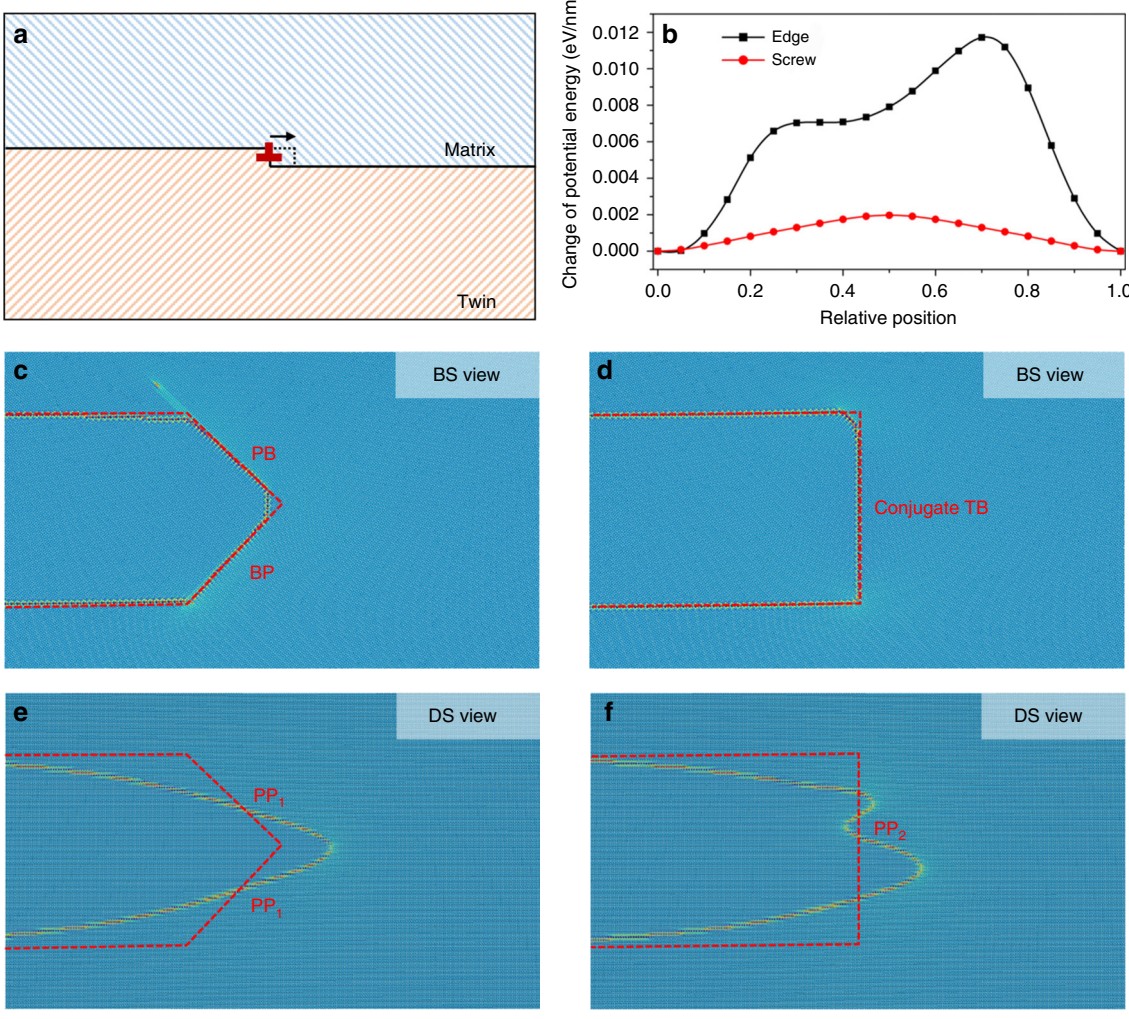

**Fig. 5** Atomistic simulations manifest faster lateral propagation after a twin nucleates. **a**, **b** The kinetic barrier of a straight edge or screw twinning dislocation (TD), indicating a higher mobility for screw TD than for edge TD. **c**, **d** Molecular Dynamics (MD) simulations showing that the facets are stable for the BS view with 100 ps relaxation at 1 k. **e**, **f** MD simulations showing that the facets degenerate into multiple two-layer steps (TDs) on the DS view. This finding suggests the strong obstruction by non-equilibrium boundaries from the BS view

view (mostly TPP boundaries) grow faster than BP/PB serrated boundaries in the BS view remain unknown.

## Discussion

The mobility of forward and lateral TBs is strongly dependent on the character of TDs and the pinning effect associated with steps/facets along the forward and lateral TBs. When a TD loop nucleates on the normal-TB, the twin thickens by two $\{0\bar{1}12\}$ atomic layers. The migration of forward and lateral TBs is accomplished via glide of TD loops[34] along $\eta_1$ and $\lambda$ directions. The segment of a TD loop parallel to $\lambda$ is perpendicular to the Burgers vector and has pure edge character. The segment of a TD loop parallel to $\eta_1$ is parallel to the Burgers vector and has pure screw character. For a general dislocation with planar core, a screw dislocation has higher mobility than an edge dislocation[24,35]. This can partly account for the observed aspect ratios of twins reported in Fig. 4. We further confirmed this by calculating the kinetic barriers of a straight edge or screw TD using molecular statics (MS) simulations.

The kinetic barriers are calculated by the nudge elastic band method[36] in which the initial and final configurations of a single step are schematically shown in Fig. 5a (construction and simulation details are described in "Methods" section). With 19 intermediate states, the kinetic barriers associated with the glide of edge and screw TDs are calculated and plotted. We find that the barrier for the glide of edge TD is around 0.012 eV/nm, which is six times that of screw TD shown in Fig. 5b. This indicates a higher mobility for screw TDs than for edge TDs, and so a faster lateral migration of twins, compared to the forward migration. In addition, recent 3D atomic simulations of twin growth[25] also show that the twin nucleus grows faster on the lateral side than on the forward side when shear is applied (see Supplementary Fig. 6).

The mobility of TDs, however, is not the only mechanism to account for the difference in TB migration, because steps/facets are often formed during propagation and growth of twins. At early stages of twinning, it is clear that lateral and forward TBs are in a state of nonequilibrium associated with the pileup of TDs. The questions to be posed next are whether large coherent or semicoherent facets form, as twins become larger; and whether those facets can move and lead to twin propagation. Molecular dynamics (MD) simulations show that edge TDs pile up and form BP/PB steps/facets in the BS view, and screw TDs pile up and form T-PP1 (semicoherent $\{\bar{1}101\}||\{10\bar{1}1\}$ interface with 5.68° twist angle) or/and T-PP2 (semicoherent $\{\bar{2}110\}||\{2\bar{1}10\}$ interface with 7.42° twist angle) facets[34]. When twinning is accomplished by the migration of large facets, shuffling is simultaneously involved. Since shuffling is a diffusion process,

 **5**

more shuffling leads to a higher energy barrier, which would reduce the mobility of large steps/facets compared to the one of individual TDs. As a consequence, although in a relaxed unstressed state large facets may be energetically favorable, our previous work reveals that propagation requires a break-up of coherency and the formation of local edge and screw TDs pile-ups on the forward and lateral sides. Here, we constructed different facets and examined the stability of these large coherent facets using MD simulations. Red dash lines in Fig. 5c show the initial shape of twin tips. (Construction and simulation details are described in "Methods" section). In Fig. 5c, d, with 100 ps relaxation at 1 k, BP/PB facets in the BS view are stable. Meanwhile, T-PP1 and T-PP2 facets in the DS view (Fig. 5e, f) degenerate into multiple separated 2-layer steps. Thus, the twin should grow faster in the DS view due to higher mobility of TDs than in the BS view associated with formation of large PB/BP facets. Formation of misfit dislocation will slow down twin propagation in both forward and lateral directions (details can be found in Supplementary Note 2 and Supplementary Fig. 7). However, quantitative estimation of the propagation that is retarded is not clear and will be the subject of future work.

In summary, by applying EBSD statistical analysis to 276 representative deformation twins in magnesium, we found that a lenticular twin shape when viewed along the twinning shear direction (the DS view, long axis along $\lambda$ and short axis along $k_1$), same as the classic shape associated with the lateral viewing (the BS view, $\eta_1$ is the long axis and $k_1$ is the short axis). This reveals that forward and lateral propagation of twins is easier than the normal growth perpendicular to the $\{10\bar{1}2\}$ CTB. The twin section has an irregular shape when observed along the $k_1$ direction (TPN view), and $\lambda$ tends to be longer than $\eta_1$, at least until grain boundaries or other twins start interfering with propagation ($\lambda < 40\,\mu m$). The latter evidence, combined with the fact that DS sections tend to fully span the grain (type 2 in Fig. 4a), implies that lateral expansion is faster than forward propagation. Also, the irregular morphology suggests small or null surface tension associated with twin interfaces. We attribute the anisotropic propagation to the different mobility of the edge and screw components of TDs. These findings improve the current understanding of deformation twin morphology and propagation in HCP metals. In addition, these findings lead us to speculate that, because both forward and lateral twin motion is required for twin growth, their relative mobility may provide an explanation to why different twin modes can (or cannot) be activated in different HCP materials. However, the effect of misfit dislocation on twin propagation in both, forward and lateral directions, suggests the need for a full 3D characterization of individual twins combining serial sectioning and 3D molecular dynamics.

## Methods

**Experimental characterization**. A commercial pure, fully recrystallized Mg plate with a strong basal texture component perpendicular to the rolling plane was compressed along the rolling direction (RD) to a total strain of 1% to activate co-zonal $\{01\bar{1}2\}$ twins[37]. The electro-polished samples were cut using two different sections: the most commonly used one, containing the RD and the normal direction (ND), to reveal the BS view of twins, and an atypical one, at 45° to the RD and the ND, to maximize the number of twins from TPN view and DS view. Samples were electropolished in a solution of 2% nitric acid and water at a voltage of less than 1 V. An FEI XL30 with an accelerating voltage of 25 kV was used for EBSD to obtain crystal orientation for both parent and twin phases. Total examined area is 8 mm long and 4 mm wide.

**Atomistic simulations**. MS/MD simulations were conducted for Mg with the empirical interatomic potential developed by Liu et al.[38].

Under the convention that $x$-axis is along $[10\bar{1}1]$ direction, $y$-axis is normal to $(\bar{1}012)$ plane and $z$-axis is along $[1\bar{2}10]$ direction, construction of the model containing an edge TD starts with a $40 \times 40 \times 1.60$ nm bicrystal with $(\bar{1}012)$ twinning orientation. After introducing a pure edge TD with Burgers vector $b_t^e = (0.049, 0, 0)$ nm by applying the anisotropic Barnett–Lothe solutions[39] followed by

shuffle operation[34], a two-layer step with pure edge TD is constructed. Enforcing fixed boundaries in $x$- and $y$-direction and periodic boundary condition in $z$-direction, dynamic quenching is conducted with the EAM potential developed by Liu et al.[38] until the maximum force is less than 5 pN. Similarly, a $40 \times 40 \times 1.52$ nm $(\bar{1}012)$ twinning-oriented bicrystal model containing a pure screw TD is constructed with coordinate $x$-axis along $[1\bar{2}10]$ direction, $y$-axis normal to $(\bar{1}012)$ plane and $z$-axis along $[10\bar{1}1]$ direction. Next, a pure screw TD with Burgers vector $b_t^s = (0, 0, 0.049)$ nm and shuffle operations are incorporated to the model. The configurations for screw disconnection are relaxed under the same boundary condition and equilibrium criterion as the case of edge disconnection.

The construction of models containing BP/PB and $k_2$ facets starts with an $80 \times 80 \times 1.60$ nm single crystal in the coordinate that $x$-axis is along $[10\bar{1}1]$ direction, $y$-axis is normal to $(\bar{1}012)$ plane, and $z$-axis is along $[1\bar{2}10]$ direction. By introducing 64 edge TDs with Burgers vector $b_t^e = (0.049, 0, 0)$ nm[39] onto BP/PB or K$_2$ planes and associated shuffle vectors[34], twin tips containing BP/PB or $k_2$ facets are constructed. Similarly, the construction of models containing T-PP1 and T-PP2 facets starts with an $80 \times 80 \times 1.52$ nm single crystal which adopts the coordinate that $x$-axis is along $[1\bar{2}10]$ direction, $y$-axis is normal to $(\bar{1}012)$ plane, and $z$-axis is along $[10\bar{1}1]$ direction. Totally, 64 screw TDs with Burgers vector $b_t^s = (0, 0, 0.049)$ nm are introduced into T-PP1 or T-PP2 facets followed by corresponding shuffle operations. With fixed boundary conditions in $x$- and $y$-direction and periodic boundary condition in $z$-direction, the models are relaxed at 1 k for 100 ps.

## Data availability

The data that support the findings of this study are available from the corresponding author upon reasonable request.

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

## Acknowledgements

This work is fully supported by the Office of Basic Energy Sciences, Project FWP 06SCPE401, under US DOE Contract no. W-7405-ENG-36.

## Author contributions

Y.L. and R.J.M. prepared the specimen and performed the microscopy experiments. Y.L. and P.Z.T. performed the statistical EBSD analysis. Atomistic simulations were conducted by M.G. and J.W. C.N.T. conceived and coordinated the entire project. All authors commented on the paper.

## Additional information

**Competing interests:** The authors declare no competing interests.

