## [Peer Review File · Nature Communications]

Reviewers' comments:

Reviewer #1 (Remarks to the Author):

In this paper the authors are trying to answer the question if the kinetics of twin growth in BS and DS directions are comparable. To do so, they characterized the twin shape from three orthogonal directions, especially in the twin plane normal direction. Their Mg plate specimen was compressed by 1%. They drew their conclusions based on their EBSD characterization of many twins in this condition. The EBSD characterization was made on the surface of their specimens. It is a 2D observation. One question to the authors is why not do some 3D EBSD work to directly establish the real shape of twins. 3D EBSD can be done quite easily nowadays. I think the best way is to use 3D EBSD to show the 3D shape of twins, rather than making deductions from 2D intersections. From the 3D EBSD, the real shape of a single twin can be worked out more accurately. As for the kinetics of twin growth, a question to the authors is why not do some in-situ experiments in SEM, and EBSD, to directly observe the propagation, growth and thickening of a twin along some specific directions. Last question to the authors: are the facets of twins shown in figure 1e supported by the EBSD maps presented in figure 2?

Reviewer #2 (Remarks to the Author):

The authors performed statistical analysis of twin shape along three viewing directions: the 'dark side' view, the twinning plane normal view and the 'bright side' view. To do so, a Mg plate was compressed to activate co-zonal {10-12} twins. EBSD analysis revealed that twins present lenticular shape in the dark side but irregular shape along the twinning plane normal view. MD simulations were performed to interpret the experimental observations of twin growth in terms of the structure of the interfaces, related twinning dislocations and their interaction with misfit dislocations on the twin boundaries.

The authors have a proved experience in the topic. This paper is directly linked to ref 33. The research is rigorous and contributes to the understanding of the most abundant twin in hcp structures, i.e. the {10-12} twin.

I have just few remarks on the content and/or wording:

Page 3 line 5:

"BP/PB facets ...atomistic simulations of Mg 29-31" I think it is illustrative to include ref.5 of supplemental information here. Probably it is more appropriate than ref. 30 to illustrate PB facets

Page 5, last 4 lines:

a) "The segment of a TD loop which is parallel to [THE BURGERS VECTOR (??)] η_1 has pure screw character".

b) "For a dislocation with planar core, a screw dislocation has higher mobility than an edge dislocation 34, 35"

- Are references 34 and 35 appropriate in the context of this paragraph that refers precisely to the TD in the {10-12} twin?

- Since the TD has step character, the meaning of 'planar core' should be clarified.

Page 7 of supplementary information:

"...the stresses can be relaxed by the addition of misfit partials with Burgers vectors (??)"

There are other typos like in the caption of figure S5: ... twin/matrix is [BE] described...

Responses to reviewer #1

Overall comment

“In this paper the authors are trying to answer the question if the kinetics of twin growth in BS and DS directions are comparable. To do so, they characterized the twin shape from three orthogonal directions, especially in the twin plane normal direction. Their Mg plate specimen was compressed by 1%. They drew their conclusions based on their EBSD characterization of many twins in this condition. The EBSD characterization was made on the surface of their specimens. It is a 2D observation.”

Answer: We thank the reviewer for the interpretation of our work. We want to highlight what we believe is the main result from our study: statistical analysis of hundreds of twin shapes in the early stages of formation allows us to infer the mechanisms of forward and lateral propagation. We conclude that in the first stage twins have a tendency to expand laterally and, past a critical ‘width’, forward propagation is favored. This result is novel and we complement it with MD simulations that justify such behavior in terms of twin-facet mobility and formation of interface dislocations. Statistical analysis plays an important role since twin shapes are neither perfect nor identical, and driving stresses are likely to vary from grain to grain. Our analysis of 2D EBSD sections has the advantage over 3D twin reconstruction that a much larger number of twins can be analyzed, and that the statistical analysis of their sections reveals more microscopic information about the shape of the twin domain. We will explain this in the following point-by-point response in further detail.

Comment #1

“One question to the authors is why not do some 3D EBSD work to directly establish the real shape of twins. 3D EBSD can be done quite easily nowadays. I think the best way is to use 3D EBSD to show the 3D shape of twins, rather than making deductions from 2D intersections. From the 3D EBSD, the real shape of a single twin can be worked out more accurately.”

Answer: The assertion that ‘3D EBSD can be done quite easily today’ is misleading. True that the software DREAM-3D is available for reconstructing sections measured with EBSD. This technique has been applied mainly to reconstruct aggregates and investigate grain geometry [T. Liu et al, 2017, 2018] and for reconstituting the shape of three ‘well developed’ twins in Mg [Fernandez et al, 2013]. However, the technique by serial sectioning with a focused ion beam (FIB)

has limitations for been applied to our problem, where early stages of twinning are addressed. Specifically, the resolution required to reconstruct incipient twins is at least 0.5 micron. At such scale, damage and heat induced by FIB introduce uncertainties larger than 1 micron [Gutierrez, 2017, and Mingard et al, 2018]. Moreover, the number of twins that can be characterized with this technique is relatively small.

In addition, there is the issue of automated twin-matrix orientation and twin features recognition (such as dimensions, inclination of habit plane with respect to the section, etc). Finally, once all twin features are collected, specific software is required for analyzing correlations. In our DOE-BES Program we are currently in the process of developing such 3D software. The bottom line is that analysis of twin structures and their correlations using FIB, 3D EBSD, and automated image recognition software is still in its infancy. Furthermore, an advantage of using a statistical approach to analyze 2D sections (in our case this was done manually via exhaustive microscopy work) is that one can obtain distributions of shapes and general trends associated with the twin-parameters studied. In addition to being extremely time intensive, 3D-EBSD reconstruction technique will not provide sufficient number of twins to overcome variabilities and derive a meaningful conclusion in this study. As an example, the statistics on twin aspect ratios shown in **Figure R1**, for twins that intersect none, one or two grain boundaries, indicate that in all three cases there is a threshold length that divides the early faster lateral growth from the later faster forward growth. The main text has been revised accordingly.

Figure R1 Statistical analysis showing that for all three types of twin shapes, they all have critical “width” to divide the early faster laterally growth from the later faster forward growth.

Reference

- A. Fernandez, A. Jérusalem, I. Gutiérrez-Urrutia, M.T. Pérez-Prado, “3D investigation of the grain boundary-twin interactions in a Mg AZ31 alloy by 3D EBSD and continuum modeling”, Acta Mater 61 (2013) 7679-92
- Mingard, K. P., Stewart, M., Gee, M. G., Vespucci, S., Trager-Cowan, C. Practical application of direct electron detectors to EBSD mapping in 2D and 3D. Ultramicroscopy 184, 242-251 (2018).
- Gutierrez-Urrutia, I. Analysis of FIB-induced damage by electron channeling contrast imaging in the SEM. J Microsc 265, 51-59 (2016).
- Liu, T., Xia, S., Zhou, B., Bai, Q., Rohrer, G. S. Three-dimensional geometrical and topological characteristics of grains in conventional and grain boundary engineered 316L stainless steel. Micron 109, 58-70 (2018).
- Liu, T., Xia, S., Zhou, B., Bai, Q., Rohrer, G. S. Three-dimensional characteristics of the grain boundary networks of conventional and grain boundary engineered 316L stainless steel. Mater Charact 133, 60-69 (2017).

Comment #2

“As for the kinetics of twin growth, a question to the authors is why not do some in-situ experiments in SEM, and EBSD, to directly observe the propagation, growth and thickening of a twin along some specific directions.”

Answer: ‘in-situ experiments in SEM, and EBSD, to directly observe the propagation, growth and thickening of a twin along some specific directions’ is, in principle, an ideal approach for characterizing twin kinetics at different stages. The authors have experience with this technique and have utilized it in the past [Liu Y. et al, 2014, 2016, 2017]. However, there are two issues (geometry constraints on twinning and dimension resolution associated with microscopy), and one realistic challenge, that preclude doing in-situ experiments in this work.

Twins can nucleate either inside or from the surface of the sample, and twin propagation and growth are subjected to elastic constraint associated with surrounding material. In-situ observation is able to capture the change in 2D morphology of twins on the observed surface, but elastic constrains on surface are quite different from those inside the bulk of the sample. For in-

situ TEM observation the sample thickness ranges from less than 100 nm up to a few hundred nanometers. In-situ TEM mechanical testing (compression and tension) of nanosized single crystals [Liu BY et al, 2014], show that under such conditions twin propagation and growth can be accomplished through different interface motion mechanisms than the dominant ones for twinning in bulk samples [Hirth JP, Wang J, Tomé CN, Prog Mater Sci 2016; Gong MY et al, 2017].

The dimension resolution of twins is related to microscopic techniques used for in-situ observation. EBSD is not applicable to characterization of kinetics because of the long scanning time for one image. Optical microscopy is good for capturing dynamic propagation and growth of twins. We have done many observations in Mg and Mg alloys [Yu Q et al, 2014a, 2014b]. However, it does not address bulk behavior and the dimension resolution is at best 2 micron, which has limitations when early stages of twinning are addressed. SEM is limited by the sample size and observed directions, and is unable to capture the three stages of twinning in bulk sample (Liu Y et al, 2017).

Lastly, the realistic challenge for doing in-situ observation is the extreme time-consuming aspect of the technique, and that it will not capture a sufficient number of twins to overcome variabilities and provide statistically meaningful conclusions.

Reference

Gong M, Hirth JP, Liu Y, Shen Y, Wang J. Interface structures and twinning mechanisms of twins in hexagonal metals. Materials Research Letters 2017, 5: 1-16.

Hirth JP, Wang J, Tomé CN. Disconnections and other defects associated with twin interfaces. Progress in Materials Science 2016, 83: 417-471.

Liu BY, Wang J, Li B, Lu L, Zhang XY, Shan ZW, et al. Twinning-like lattice reorientation without a crystallographic twinning plane. Nature Communications 2014, 5(2): 3297.

Liu Y, Karaman I, Wang H, Zhang X. Two Types of Martensitic Phase Transformations in Magnetic Shape Memory Alloys by In-Situ Nanoindentation Studies, vol. 26, 2014.

Liu Y, Li N, Bufford D, Lee JH, Wang J, Wang H, et al. In Situ Nanoindentation Studies on Detwinning and Work Hardening in Nanotwinned Monolithic Metals. JOM 2016, 68(1): 127-135.

Liu Y, Li N, Kumar MA, Pathak S, Wang J, McCabe RJ, et al. Experimentally quantifying critical stresses associated with basal slip and twinning in magnesium using micropillars. Acta Materialia 2017, 135: 411-421.

Yu Q, Wang J, Jiang Y, McCabe RJ, Tomé CN. Co-zone \${10\bar{1}2}\$ Twin Interaction in Magnesium Single Crystal. Materials Research Letters 2014, 2(2): 82-88.

Yu Q, Wang J, Jiang Y, McCabe RJ, Li N, Tomé CN. Twin–twin interactions in magnesium. Acta Materialia 2014, 77: 28-42.

Comment #3

“Last question to the authors: are the facets of twins shown in figure 1e supported by the EBSD maps presented in figure 2?”

Answer: Because of the resolution limitation of EBSD techniques, the answer is definitely ‘no’, even when it is possible to observe some very sharp, straight boundary ‘traces’. It must be noted that the ‘trace’ represents the intersection of the boundary plane with the observed surface, and there is no way to confirm whether the boundary plane is perpendicular to the surface. Thus, the boundary outlined in EBSD images is a cross-section of the twin. More importantly, the length resolution of EBSD techniques is limited to about tens of nanometers. Except coherent twin boundary which could be flat with a micrometer dimension, the dimension of other facets that form associated with the pileup of twinning dislocations, could be much smaller than the resolution limit of EBSD.

Responses to reviewer #2

Overall comment

“The authors performed statistical analysis of twin shape along three viewing directions: the ‘dark side’ view, the twinning plane normal view and the ‘bright side’ view. To do so, a Mg plate was compressed to activate co-zonal {10-12} twins. EBSD analysis revealed that twins present lenticular shape in the dark side but irregular shape along the twinning plane normal view. MD simulations were performed to interpret the experimental observations of twin growth in terms of the structure of the interfaces, related twinning dislocations and their interaction with misfit dislocations on the twin boundaries.

The authors have a proved experience in the topic. This paper is directly linked to ref 33. The research is rigorous and contributes to the understanding of the most abundant twin in hcp structures, i.e. the $\{10\bar{1}2\}$ twin.”

Answer: We thank the reviewer to stand by the quality of our work and the rigor of the conclusions.

Minor Comments

(1) “Page 3 line 5: “BP/PB facets ...atomistic simulations of Mg 29-31” I think it is illustrative to include ref.5 of supplemental information here. Probably it is more appropriate than ref. 30 to illustrate PB facets”

Answer: We thank the reviewer’s comments. The ref 30 has been replaced with “Barrett CD, Kadiri HE, The roles of grain boundary dislocations and disclinations in the nucleation of $\{10\bar{1}2\}$ twinning, Acta Materialia 2014, 63: 1-15.”

(2) “Page 5, last 4 lines:

- a) The segment of a TD loop which is parallel to [THE BURGERS VECTOR (??)] η_1 has pure screw character.
- b) For a dislocation with planar core, a screw dislocation has higher mobility than an edge dislocation 34, 35
 - “Are references 34 and 35 appropriate in the context of this paragraph that refers precisely to the TD in the {10-12} twin?”
 - “Since the TD has step character, the meaning of ‘planar core’ should be clarified.”

Answer:

(a) A revision has been made accordingly, as “The segment of a TD loop which is parallel to the Burgers vector is also parallel to η_1 and has pure screw character”

(b) A revision has been made accordingly, as “For a general dislocation with planar core, a screw dislocation has higher mobility than an edge dislocation”. Also, Ref 35 has been replaced with Ref 24.

(3) Page 7 of supplementary information:

- the stresses can be relaxed by the addition of misfit partials with Burgers vectors (??)”
- There are other typos like in the caption of figure S5: ... twin/matrix is [BE] described

Answer: We thank the reviewer to point out these typos. A revision has been made accordingly.

Reviewers' comments:

Reviewer #1 (Remarks to the Author):

Even if the authors feel that 3D EBSD is beyond their capability, they should at least try to analyse their 2D data points by taking the truncation effect into account. The values of twin dimensions that they provided in several figures will change when the truncation correction is made, or when the 3D approach is used.

The essence of this paper is the statement made by the authors: a deformation twin grows faster in the DS direction than the BS direction in the initial stages of twin growth. What is the definition for initial stages of twin growth? For the twins that were shown in their EBSD maps, they are already several microns or even tens of microns. These sizes are far beyond initial stages of twin growth, they are also far beyond the sizes of twins in their MD simulations. The definition for the "initial stages" is quite arbitrary, so is the critical size at which the preference of growth occurs. These vague definitions do not help reader to fully understand their work.

The expectation from the classical theory is that growth along the shear direction (DS in authors' terminology) should be faster, unless the growth front develops facets of much lower interfacial energy. It would add value to this work if the authors could provide some experimental information on the structures of the facets of twins before and after the growth preference is changed.

Reviewer's comments

Reviewer #1 (Remarks to the Author):

Even if the authors feel that 3D EBSD is beyond their capability, they should at least try to analyse their 2D data points by taking the truncation effect into account. The values of twin dimensions that they provided in several figures will change when the truncation correction is made, or when the 3D approach is used.

The essence of this paper is the statement made by the authors: a deformation twin grows faster in the DS direction than the BS direction in the initial stages of twin growth. What is the definition for initial stages of twin growth? For the twins that were shown in their EBSD maps, they are already several microns or even tens of microns. These sizes are far beyond initial stages of twin growth, they are also far beyond the sizes of twins in their MD simulations. The definition for the “initial stages” is quite arbitrary, so is the critical size at which the preference of growth occurs. These vague definitions do not help reader to fully understand their work.

The expectation from the classical theory is that growth along the shear direction (DS in authors' terminology) should be faster, unless the growth front develops facets of much lower interfacial energy. It would add value to this work if the authors could provide some experimental information on the structures of the facets of twins before and after the growth preference is changed.

Response to reviewer

Overall response:

The authors appreciate the comprehensive feedback and challenges provided by the reviewer. As a result of it we have added new experimental characterization, revised our interpretation of results and some of our conclusions, and eliminated parts that were deemed more speculative. We believe that the revised manuscript is clearer than before and more impactful.

The most relevant and novel aspect of this paper is the characterization of twin morphology by analyzing non-standard sections, namely, viewing them along the twin propagation direction and along the twin plane normal. In addition, we added an analysis of the typical twin sections showing the twin profile along the coherent twin plane and the forward propagation direction.

This is an imminently experimental paper. The statistical analysis allows us to infer implications on the anisotropy and 3D mobility of twins.

Comments #1

1. Even if the authors feel that 3D EBSD is beyond their capability, they should at least try to analyse their 2D data points by taking the truncation effect into account. The values of twin dimensions that they provided in several figures will change when the truncation correction is made, or when the 3D approach is used.

Answer: If by ‘truncation effect’ the reviewer refers to sectioning the twin with a plane non-parallel to the coherent twin boundary (what we call TPN view) or not perpendicular to the CTB (DS view), the answer is: we establish the relative orientation between the twin planes and the section plane and do not consider twins that exceed 5 degree misorientation. The procedure is explained in main text (page 4) and **Fig. 3** (here is **Fig. R1**). In addition, when measuring the main directions along long axes λ and η_1 , and along short axis along k_1 , we correct the length accordingly. A schematic of BS section is shown below.

Figure R1 (Figure 3). (a) A deformation twin T_1 (pink) from BS view and (b) corresponding pole figures of the twin and matrix. The pole figures show deviation angles $\alpha = 4^\circ$ from $\{2\bar{1}\bar{1}0\}$ and $\theta = 86^\circ$ from $\{40\bar{4}7\}$ (plane normal to $\langle 10\bar{1}1 \rangle$) pole figures. (c-d) Illustration of the projection plane on EBSD sections and the measured length correction along η_1 based on the deviation angles quantified by pole figures.

On the other hand, if the reviewer refers to ‘where’ the cut intersects the twin, we present below an explanation. We classified twins in three types based on their junctions at GBs as they appear in the EBSD sections. They are Type 0 if neither side of the twin is connected to GBs, Type 1 if only one side is connected to a GB and Type 2 if both sides are connected to GBs. Since twins in annealed HCP nearly always propagate from grain boundaries, the schematic below attempts to capture such situation as shown in **Fig. R2**: (a) represents our typical picture of forward propagation after one twin nucleation, that showing type 1 case from BS view (k_1 and η_1) and type 0 case from DS view (k_1 and λ); (b) represents another scenario when twin propagation start laterally towards λ , that showing type 0 case from BS view (k_1 and η_1) and type 1 case from DS view (k_1 and λ).

Moreover, in our main text (page 4), we found: “As shown in **Fig. 4a**, the number of BS twins (η_1-k_1) that qualify is relevant for the 3 types, with prevalence of type 2; twins in DS view ($\lambda-k_1$) are overwhelmingly type 2, and practically zero number of type 0 is observed; in what concerns TPN twins, they are exclusively type 1 and type 2 in about equal numbers, indicating that

propagation always start at a GB.” This evidence suggests that the lateral expansion of twins is relatively easy, since most of the DS view ($\lambda-k_1$) extend from side to side of the grain (type 2), compared to the BS view (η_1-k_1).

Figure R2. (a) A schematic showing our typical picture of forward propagation after one twin nucleation. It manifests type 1 case from BS view (k_1 and η_1) and type 0 case from DS view (k_1 and λ); (b) A schematic showing different scenario when twin propagation start laterally towards λ . It manifests type 0 case from BS view (k_1 and η_1) and type 1 case from DS view (k_1 and λ).

Comments #2

2. The essence of this paper is the statement made by the authors: a deformation twin grows faster in the DS direction than the BS direction in the initial stages of twin growth. What is the definition for initial stages of twin growth? For the twins that were shown in their EBSD maps, they are already several microns or even tens of microns. These sizes are far beyond initial stages of twin growth, they are also far beyond the sizes of twins in their MD simulations. The definition for the “initial stages” is quite arbitrary, so is the critical size at which the preference of growth occurs. These vague definitions do not help reader to fully understand their work.

Answer: We agree with the reviewer that ‘initial stage’ may convey different length scales to different readers. We have added clarifications in the introduction:

“Twinning involves three sequential processes: nucleation, propagation and growth, which are associated with formation and migration of twin boundaries (TBs). This work focuses on the twin morphology at the propagation stage, defined as the twin having partially or fully traversed the grain, but before it starts growing in thickness after being arrested at the opposite grain boundary”, and

“Experimental results also show that twins grow faster in the direction λ (associated with migration of lateral TBs) than in the direction η_1 (associated with migration of forward TBs) at the initial stages of twin growth, before intersection with grain boundaries that arrest the forward and lateral propagation”.

Comment #3

The expectation from the classical theory is that growth along the shear direction (DS view in authors' terminology) should be faster, unless the growth front develops facets of much lower interfacial energy. It would add value to this work if the authors could provide some experimental information on the structures of the facets of twins before and after the growth preference is changed.

First a clarification: in our nomenclature the propagation along the shear direction would be apparent in the Bright Side (BS) view of the twin.

Answer:

We thank the reviewer's comment. As shown in Fig. R5, we provide additional facet analysis that may suggesting possible to form $\{42\bar{6}7\}$ facet from TPN view (Fig.1e)

Figure R5. The EBSD maps showing the possible facets as shown in Figure 1e.

We are not aware of a classical theory result about the 3D motion of twins. Classical crack or twin analysis tends to look always at the 2D forward stress field, and in a few of those calculations (mostly finite element method) also to propagate. Even in such 2D case the problem is complex because dislocation emission relaxes the forward stress, and the analysis should include plasticity.

We do not recall the lateral stress field and associated relaxation being discussed in the literature. As a consequence, we asked one of our collaborators (Dr. M. Arul Kumar) to calculate the 3D stress field around a flat twin domain (10:10:1 aspect ratio). He used the Fast Fourier Transform approach described in [M. Arul Kumar, A.K. Kanjarla, S.R. Niezgod, R.A. Lebensohn, C.N. Tomé, “Numerical study of the stress state of a deformation twin in magnesium”, *Acta Materialia* 84 (2015) 349-358] but extended it to 3D. He simulates the twin shear transformation under the action of an applied shear stress on the twin plane. Stress relaxation in the matrix surrounding the twin is accounted for by allowing for basal $\langle a \rangle$ and prism $\langle a \rangle$ slip. The resolved shear on the twin plane (plotted below), is positive on the forward and the lateral side of the twin (**Fig. R3**). Allowing for plasticity makes a qualitative difference in the calculated stress field. A profile of Resolved Shear along the twin direction is included below, plot along forward and laterally oriented lines. Observe that inside the twin there is a negative back-shear induced by the twin transformation, but the positive shear along the rim will favor both, edge forward twin dislocations and screw lateral twin dislocations. Since the forward rim and lateral rim shear has about the same value, and since screw TDs have a lower activation barrier, it is to be expected that this will favor the lateral expansion of the twin. This result can be regarded as valid for a continuum micrometers scale. In what concerns the atomic scale, we provide simulation results in **Supplementary Fig. S6** which also show a rapid lateral propagation of the twin (**Fig. R4**).

Figure R3. Fast Fourier Transform approach described twin shear transformation under the action of an applied shear stress on the twin plane. Stress relaxation in the matrix surrounding the twin is accounted for by allowing for basal $\langle a \rangle$ and prism $\langle a \rangle$ slip. The resolved shear on the twin plane (plotted below), is positive on the forward and the lateral side of the twin

Figure R4 (Supplementary Figure S6) (a) 3D atomic configuration of the twin nucleus. (b) Under loading, the twin nucleus grows faster on the lateral side than on the forward side.

In sum, the reviewer rightly questions the idea of a transition, and requests experimental evidence of micron sized facets of twins (which we do not have). We revised our speculation that there is a transition from lateral to forward propagation based on a change in mechanisms. At the scale of the twins that we measure (well over one micron, as the reviewer points out) it is unlikely. Instead, we attribute the transition to the fact that the expanding twin is arrested by barriers that have a separation close to the ~40-micron transition revealed by **Fig. 4**. The path between barriers is consistent with the grain size or other twins in the grain. We added such argument in the analysis of **Fig. 4**.

“Such behavior holds up to $\lambda \sim 40 \mu\text{m}$, past this length, the length η_1 appears to be systematically larger than λ . However, the scarcity and dispersion of the results does not guarantee a firm conclusion, and we speculate that at this point the propagation of the expanding twin is affected by grain boundaries or by other twins.”

REVIEWERS' COMMENTS:

Reviewer #1 (Remarks to the Author):

The revised manuscript is now easier to read. It is publishable after the correction of the following grammar issues by the typesetter:

1. Page 5, "Such behavior holds up to $\lambda \sim 40 \mu\text{m}$, past this length," should be "Such behavior holds up to $\lambda \sim 40 \mu\text{m}$, above which".
2. Page 7, top paragraph, line 3, "do large coherent or semi-coherent facets form as twins become larger?", is this a sentence?
3. Page 7, top paragraph, "Formation of misfit dislocation will slower twin propagation", will slower??
4. Page 8, "may suggests" should be "may suggest".
5. Page 11, figure 1 caption, "The longest length of each axis in each section were used" should be "The longest length of each axis in each section was used".
6. Page 15, "MD simulations showing the facets are" should be "MD simulations showing that the facets are".

Response to reviewer's comments (Manuscript No. NCOMMS-18-13381156B).

The revised manuscript is now easier to read. It is publishable after the correction of the following grammar issues by the typesetter:

1. Page 5, "Such behavior holds up to $\lambda \sim 40 \mu\text{m}$, past this length," should be "Such behavior holds up to $\lambda \sim 40 \mu\text{m}$, above which".

Answer: This sentence has been revised and marked in yellow.

2. Page 7, top paragraph, line 3, "do large coherent or semi-coherent facets form as twins become larger?", is this a sentence?

Answer: This sentence has been revised as: "...whether large coherent or semi-coherent facets form, as twins become larger...".

3. Page 7, top paragraph, "Formation of misfit dislocation will slower twin propagation", will slower??

Answer: This sentence has been revised as: "Formation of misfit dislocation will slow down the twin propagation".

4. Page 8, "may suggests" should be "may suggest".

5. Page 11, figure 2 caption, "The longest length of each axis in each section were used" should be "The longest length of each axis in each section was used".

6. Page 15, "MD simulations showing the facets are" should be "MD simulations showing that the facets are".

Answer: These three comments have been revised accordingly and marked in yellow.